# Mothers' perceptions of universal newborn hearing screening in transitional Albania

**Birkena Qirjazi** [1,2]☯, **Ervin Toçi** [1,2]☯ *, **Eduard Tushe** [3‡], **Genc Burazeri** [1‡], **Enver Roshi** [1‡]

**1** Faculty of Medicine, University of Medicine, Tirana, Tirana, Albania, **2** Institute of Public Health, Tirana, Albania, **3** University Maternity Hospital "Koço Gliozheni", Tirana, Albania

☯ These authors contributed equally to this work.
‡ These authors also contributed equally to this work.
* ervintoci@yahoo.com

**Data Availability Statement:** All relevant data are within the manuscript and its Supporting Information file.

**Funding:** This project has received funding from the European Union's Horizon 2020 research and

## Abstract

The universal newborn hearing screening (UNHS) component of the multi-center EUSCREEN project is being piloted in Albania since January 1st 2018. The aim of this study was to explore mothers' perceptions about various elements of UNHS in Albania. A cross-sectional study was carried out in the three sites of UNHS in Albania, namely in Tirana, Kukës and Pogradec during May-June 2019. During this period 512 consecutively approached mothers giving birth to included maternity hospitals were interviewed face-to-face about different aspects of UNHS. Basic socio-demographic and socioeconomic information was also collected. Mean age of participating mothers was 28.6 years ± 5.5 years. The overwhelming majority (93%) of mothers knew what their baby was being tested for, 33% were aware that hearing screening was offered in maternity hospital, 94% were very satisfied/satisfied with UNHS and about 62% were very stressed/stressed waiting for screening results, with significant sociodemographic and socioeconomic differences. The main information source about UNHS was screening staff in the maternity hospitals where mothers gave birth, reported in 67% of cases. All mothers (100%) agreed on the importance of early detection of newborn hearing problems, all mothers were willing to be informed early if their newborn baby had a hearing problem and all mothers were willing to contribute financially for testing the hearing of their newborn baby. These findings should guide information and education campaigns about UNHS in Albania. The public willingness to financially support neonatal hearing testing should be considered as an opportunity to achieve universal newborn hearing screening in the country.

## Introduction

Screening newborns for hearing impairment or hearing loss is important as this is one of the most frequent congenital conditions, affecting about 1–6 newborns per 1000 live births [1,2] with much higher rates among high-risk infants admitted to neonatal intensive care unit (NICU) [3]. In addition, about half of hearing impaired newborns have no known risk factors [1] highlighting the necessity to screen all newborns. Furthermore, early detection and

innovation program under grant agreement No 733352. This study sponsor had no role in the study design, or the collection, analysis and interpretation of the data or in the writing of the report or in the decision to submit the article for publication.

**Competing interests:** The authors have declared that no competing interests exist

treatment of hearing loss is important for ensuring the normal development of speech and language and avoiding negative effects on cognitive, academic and social-emotional skills of affected children [4].

Even though universal newborn hearing screening (UNHS) is mainly implemented in developed countries [5], the practice is expanding in developing countries as well [6]. The main challenge with UNHS in developed countries is the low rate of screening follow-up. For example, in USA in 2012 about one third of newborns that did not pass the first hearing screening test failed to receive timely follow-up [7]. On the other hand, implementation of UNHS in developing countries faces challenges of different natures, including limited coverage, high unaffordable cost, lack of health personnel, poor infrastructure, poor access to rural areas, socio-economic differences, etc. [8].

Besides these factors, people's perceptions of health services constitutes an important determinant of utilization of available health services and a key element for ensuring that health interventions and services are successful [9]. Various factors contribute to shape one's perceptions about health services being offered in a certain area including culture, traditions and different inherent characteristics of the health system/facility, the quality of interactions with the health personnel, etc. [9] with health staff behavior potentially moderating the effect of health services on patient satisfaction level [10].

There is a direct link between perception of a health service and users' satisfaction. Research shows that the perception about the quality of a health service is strongly and directly linked with the satisfaction users get from that service and the future use of such services [11].

In this context, parents' perception about newborn hearing screening could play an important role in improving screening program participation rates, decrease of the refusal rates and ensure that more newborns will be tested as well [12].

In Albania, a small country in South East Europe, the EUSCREEN project, universal newborn hearing screening component, started to be implemented on January 1st 2018 on three country sites and namely in Tirana, the capital city of Albania, Kukës and Pogradec municipalities [13]. Parents' opinion on certain aspects of hearing screening is important and the "acceptability" of the screening instrument by the target population is one of the criteria to be taken into consideration when applying a particular screening procedure to a community [14]. Therefore, in the context of the EUSCREEN project, a survey was carried out in order to shed light on mothers' perceptions of various aspects of universal newborn hearing screening in Albania.

## Methodology

### Type of study

This is a cross-sectional study conducted during May-June 2019.

### Population study and sampling

The target population of the study comprised the mothers of newborn babies in the maternity hospitals where newborn hearing screening is being implemented in Albania, and more specifically in Tirana, Kukës and Pogradec.

Since there are approximately a total of 11500 births each year in these 3 districts (about 80% in Tirana) and due to the inability to include the entire birth cohort in the study over a year, we decided to draw a sample of mothers who gave birth to infants in these districts' maternity hospitals. WINPEPI statistical program was used to calculate the sample size based on the following parameters: alpha error = 5%, study power = 80%, assumed proportion of mothers in favor of hearing screening = 50% (this level enables maximization of sample size);

acceptable difference = 4.3%; source population = 11500. The application of these parameters generated a sample size of 497 individuals.

In total, 512 mothers were interviewed (200 at Koco Gliozheni Maternity Hospital in Tirana, 200 at Queen Geraldine Maternity Hospital in Tirana as well, 57 at Kukës Maternity Hospital and 55 at Pogradec Maternity Hospital). It should be noted that among the 400 births included in the two maternity hospitals in Tirana, in 76 of cases the mothers did not reside in Tirana but in other districts, as many future mothers choose to give birth in one of the Tirana maternities. Mothers who gave birth to babies in Tirana but did not reside in Tirana lived in a large variety of cities and other areas of Albania, from south to north.

## Data collection and data collection instrument

In each of the maternities included in the study, the survey instrument (questionnaire) was applied consecutively to the mothers who gave birth to one or more living children in these maternity hospitals during May-June 2019, until the preset number of questionnaires (at least) was reached.

Information on mothers' opinions on the acceptability of newborn hearing screening was collected through a face-to-face administered questionnaire.

The questionnaire contained key socio-demographic data of mothers, such as: residence (urban vs. rural); age, marital status, education level; employment status, social class, economic situation, and overall health status.

The next section of the questionnaire contained specific questions regarding the acceptability of newborn hearing screening. Questions covered issues such as: being aware that the baby is being tested for hearing; being informed on the fact that screening is provided at this maternity hospital and the sources of this information; mothers' impression of hearing screening process (on a 1–5 scale where 1 means "very satisfactory" and 5 "not at all satisfactory"); the level of stress or anxiety they experienced while waiting for the hearing screening response (on a 1–5 scale where 1 means "extremely" and 5 means "not at all"); mothers' preference to be informed as soon as their child has hearing problems (yes vs. no); importance of newborn hearing screening to parents (yes vs. no); willingness to pay themselves for screening their newborn baby (yes vs. no).

The questionnaire used for this survey has been built from the local experts (the authors of this paper), based on browsing of the appropriate literature listed in the references and picking up those items considered as relevant for the study objectives. Therefore, the questionnaire has face validity, construct validity and content validity.

## Ethical issues

All mothers included in this study were approached by EUSCREEN staff screening newborns' hearing at respective maternity hospitals.

In order to be part of newborn hearing screening, each mother was required to give a written informed consent. If such consent was not retrieved, then the newborn was not screened, the mother's interviewing was not done and this particular baby was not part of UNHS anymore.

All mothers accepting to screen their baby were required to give a formal oral consent for participating in the actual study.

All mothers agreed to participate after being thoroughly informed about the survey.

EUSCREEN project in Albania has been approved by the project counterpart in Albania, i.e. the University of Medicine, and the Ministry of Health and Social Protection of Albania.

## Data analysis

Absolute numbers and the corresponding percentages were reported, in order to describe the data. Chi-square test was used to detect any statistically significant difference when comparing categorical variables.

All data analyzes were performed using the Statistical Package for Social Sciences (SPSS) statistical software, version 19.

## Results

### General information

A total of 512 mothers who had a live birth in one of the four selected maternity hospitals participated in the survey. The maternal mean age was 28.6 years ± 5.5 years, ranging from a minimum of 16 years (1 case) to a maximum of 47 years (1 case). Table 1 shows the general characteristics of participating mothers by birth survey site. More than one fourth (27.3%) of mothers were between 16 and 25 years old, about one quarter (24.4%) had secondary education or less, 56% resided in urban areas, 98% were married at the time of the survey, about half (50.2%) were employed, 91.2% declared to belong to middle social class, 80.9% had average economic status and 92.4% declared to have very good or good general health. Significantly higher proportions of mothers in Tirana had university education, resided in urban areas and had very good or good economic status, compared to mothers from other survey sites (Table 1). On the other hand, the proportion of employment was significantly higher among Pogradec mothers whereas the proportion of mothers with very good or good health status was higher among Kukës and Pogradec mothers compared to Tirana (Table 1). There were no significant age, marital status and social class differences by survey site (Table 1).

### Mother's perceptions about various aspects of newborn hearing screening

Table 2 shows mothers' perceptions about different aspects of newborn hearing screening. More than 9 in 10 mothers (93%) knew what their baby was being tested for, only one third (33%) were informed beforehand that newborn hearing screening was being offered at the respective maternity hospital where they gave birth, 94% were very satisfied or satisfied with newborn hearing screening and about 62% thought that waiting for the hearing screening result was very stressful or stressful (Table 2).

Significantly higher proportions of highly educated, employed, high social class, very good or good economic or health status mothers knew what their newborn baby was being tested for compared to their respective counterparts (Table 2).

Significantly higher proportions of mothers in Kukës (70.2%) and Pogradec (87.3%) were informed that hearing screening was offered in the maternity hospital where they gave birth compared to Tirana mothers (20.3%) and such proportion was higher among rural, high school and university, low and middle social class mothers and those with very good/good self-reported health status compared to their respective counterparts.

Significantly higher proportions of mothers in Pogradec and Kukës, and those belonging to higher social class and economic status were very satisfied/satisfied with the newborn hearing screening.

Lastly, waiting for the results of newborn hearing screening was very stressful/stressful for significantly higher proportions of Kukës mothers, those belonging to high social class and those experiencing poor/very poor health status (Table 2).

**Table 1. General characteristics of participating mothers.**

| Variable | Total N = 512 | Survey site | | | P-value |
|---|---|---|---|---|---|
| | | Tirana n = 400 | Kukës n = 57 | Pogradec n = 55 | |
| **Age-group** | | | | | |
| 16–25 years | 140 (27.3)* | 100 (25.0) | 21 (36.8) | 19 (34.5) | 0.088 † |
| 26–30 years | 193 (37.7) | 149 (37.3) | 21 (36.8) | 21 (38.2) | |
| >30 year | 179 (35.0) | 151 (37.8) | 13 (22.8) | 15 (27.3) | |
| **Education level** | | | | | |
| Secondary or les | 125 (24.4) | 105 (26.3) | 11 (19.3) | 9 (16.4) | **<0.001** |
| High school | 203 (39.6) | 131 (32.8) | 36 (63.2) | 36 (65.5) | |
| University | 184 (35.9) | 164 (41.0) | 10 (17.5) | 10 (18.2) | |
| **Residence** | | | | | |
| Urban | 288 (56.3) | 256 (64.0) | 15 (26.3) | 17 (30.9) | **<0.001** |
| Rural | 224 (43.8) | 144 (36.0) | 42 (73.7) | 38 (69.1) | |
| **Marital status** | | | | | |
| Single †† | 10 (2.0 | 6 (1.5) | 2 (3.6) | 2 (3.6) | 0.375 |
| Married | 502 (98.0) | 394 (98.5) | 55 (96.4) | 53 (96.4) | |
| **Employment status** | | | | | |
| Employed | 257 (50.2) | 202 (50.5) | 20 (35.1) | 35 (63.6) | **0.013** |
| Not employed | 245 (47.9) | 190 (47.5) | 37 (64.9) | 18 (32.7) | |
| Student | 10 (2.0) | 8 (2.0) | 0 (0.0) | 2 (3.6) | |
| **Social class** | | | | | |
| Low | 20 (3.9) | 18 (4.5) | 1 (1.8) | 1 (1.8) | 0.131 |
| Middle | 467 (91.2) | 358 (89.5) | 55 (96.5) | 54 (98.2) | |
| High | 25 (4.9) | 24 (6.0) | 1 (1.8) | 0 (0.0) | |
| **Economic status** | | | | | |
| Very good, good | 82 (16.0) | 81 (20.3) | 1 (1.8) | 0 (0.0) | **<0.001** |
| Average | 414 (80.9) | 307 (76.8) | 53 (93.0) | 54 (98.2) | |
| Poor, very poor | 16 (3.1) | 12 (3.0) | 3 (5.3) | 1 (1.8) | |
| **General health status** | | | | | |
| Very good, goo | 473 (92.4) | 361 (90.3) | 57 (100.0) | 55 (100.0) | **0.019** |
| Average | 37 (7.2) | 37 (9.3) | 0 (0.0) | 0 (0.0) | |
| Poor, very poor | 2 (0.4) | 2 (0.5) | 0 (0.0) | 0 (0.0) | |

* Absolute number and column percentage.

† P-value according to chi-square test.

†† Including divorced and widowed.

## Source of information about newborn hearing screening

Mothers reporting to have been informed that newborn hearing screening is offered in the maternity where they gave birth (n = 169 or 33% of participating mothers, see Table 2) were asked to mention the source of such information (Table 3). The majority of informed mothers (67.3%) were informed that newborn hearing screening was offered at this maternity by the medical staff at the time they were admitted to the maternity hospital, followed by information from family members, relatives or friends (in 8.6% of cases), local television programs talking about newborn hearing screening (4.9%), Internet (3.7%), posters in maternity hospital premises (3.7%) whereas other sources of information were less frequent.

**Table 2. Mothers' perception about various aspects of newborn hearing screening.**

| | Knowing what the newborn is being tested for, n (%) | Informed that hearing screening is offered in this maternity, n (%) | Very satisfied or satisfied with newborn hearing screening, n (%) | Waiting for screening result is very stressful or stressful, n (%) |
|---|---|---|---|---|
| *Total* | *476 (93.0)* | *169 (33.0)* | *482 (94.1)* | *315 (61.5)* |
| **Survey site** | NS | *** | * | *** |
| Tirana | 369 (92.3) | 81 (20.3) | 371 (92.8) | 235 (58.8) |
| Kukës | 57 (100.0) | 40 (70.2) | 56 (98.2) | 48 (84.2) |
| Pogradec | 50 (90.9) | 48 (87.3) | 55 (100.0) | 32 (58.2) |
| **Age-group** | NS | NS | NS | NS |
| 16–25 years | 125 (89.3) | 47 (33.6) | 127 (90.7) | 95 (67.9) |
| 26–30 years | 182 (94.3) | 72 (37.3) | 185 (95.9) | 118 (61.1) |
| >30 year | 169 (94.4) | 50 (27.9) | 170 (95.0) | 102 (57.0) |
| **Education level** | *** | ** | NS | NS |
| Secondary or les | 101 (80.8) | 31 (24.8) | 114 (91.2) | 74 (59.2) |
| High school | 192 (94.6) | 86 (42.4) | 194 (95.6) | 134 (66.0) |
| University | 183 (99.5) | 52 (28.3) | 174 (94.6) | 107 (58.2) |
| **Residence** | NS | * | NS | NS |
| Urban | 269 (93.4) | 79 (27.4) | 267 (92.7) | 172 (59.7) |
| Rural | 207 (92.4) | 90 (40.2) | 215 (96.0) | 143 (63.8) |
| **Marital status** | NS | NS | NS | NS |
| Single †† | 8 (80.0) | 5 (50.0) | 9 (90.0) | 6 (60.0) |
| Married | 468 (93.2) | 164 (32.7) | 473 (94.2) | 309 (61.6) |
| **Employment status** | * | NS | NS | NS |
| Employed | 246 (95.7) | 88 (34.2) | 241 (93.8) | 159 (61.9) |
| Not employed | 220 (89.8) | 78 (31.8) | 231 (94.3) | 152 (62.0) |
| Student | 10 (100.0) | 3 (30.0) | 10 (100.0) | 4 (40.0) |
| **Social class** | *** | * | ** | ** |
| Low | 11 (55.5) | 7 (35.0) | 15 (75.0) | 11 (55.0) |
| Middle | 440 (94.2) | 160 (34.3) | 442 (94.6) | 280 (60.0) |
| High | 25 (100.0) | 2 (8.0) | 25 (100.0) | 24 (96.0) |
| **Economic status** | *** | NS | *** | NS |
| Very good, good | 79 (96.3) | 19 (23.2) | 77 (93.9) | 45 (54.9) |
| Average | 390 (94.2) | 146 (35.3) | 395 (95.4) | 259 (62.6) |
| Poor, very poor | 7 (43.8) | 4 (25.0) | 10 (62.5) | 11 (68.8) |
| **General health status** | * | ** | NS | ** |
| Very good, goo | 439 (92.8) | 166 (35.1) | 444 (93.9) | 282 (59.6) |
| Average | 36 (97.3) | 3 (8.1) | 36 (97.3) | 31 (83.8) |
| Poor, very poor | 1 (50.0) | 0 (0.0) | 2 (100.0) | 2 (100.0) |

NS–not significant.

* P<0.05

** P<0.01

*** P<0.001 (P-value according to chi-square test).

†† Including divorced and widowed.

## Other aspects related to the newborn hearing screening

Participating mothers were also asked the following questions: "If your child has hearing problems, would you prefer to know this early?"; "Do you think it's important for all babies to be

**Table 3. Source of information among mothers who know that newborn hearing screening is provided at the maternity hospital where they gave birth.**

| Information source | Absolute number | Percentage |
|---|---|---|
| Various relatives or friends | 14 | 8.6 |
| Internet | 6 | 3.7 |
| Work colleagues who had previously giving birth in this maternity hospital | 1 | 0.6 |
| From previous birth | 2 | 1.2 |
| At the women's consultory | 5 | 3.1 |
| In the hospital when I was hospitalized (from staff working here) | 109 | 67.3 |
| Poster placed in maternity hospital premises | 6 | 3.7 |
| Relatives who work at the maternity hospital | 3 | 1.9 |
| Relatives undergoing hearing test in maternity | 2 | 1.2 |
| Relatives previously giving birth | 8 | 4.9 |
| Television (in general) | 3 | 1.9 |
| TV at the maternity waiting hall | 3 | 1.9 |
| **Total** | **162**[*] | **100.0** |

[*] Any discrepancy with the total number (in this case n = 169) is due to missing information.

tested for hearing in the early days of life?" and "Would you be willing to pay for a hearing test?" Analysis of their responses revealed that all mothers (100%) included in the study answered "Yes" to each of these questions (Table 4).

## Discussion

This study generated novel and interesting information that sheds light on various aspects related to the acceptability of universal newborn hearing screening program, an activity being piloted in three regions of Albania, in the framework of the EUSCREEN project.

The overwhelming majority (93%) of mothers who have given birth to children in the maternities included in the EUSCREEN project in Albania were aware of what their baby was being tested for. This suggests that the staff involved in hearing screening in all 4 participating maternity hospitals has adequately and appropriately explained to mothers the details of hearing test.

Knowledge of what the baby was being screened for was significantly and positively related to educational level, social class, economic status and overall health status. This finding has important implications for the future, suggesting that hearing screening explanation and education may need to be adapted to various educational, social, economic, and health groups for greater effectiveness.

Two-thirds of mothers who gave birth to babies in the maternities included in the EUSCREEN project in Albania were not informed that hearing screening was provided at these maternities. This finding is similar to previous research in developing countries. For

**Table 4. Opinions related to other aspects of newborn hearing screening.**

| Question | Absolute number | Percentage |
|---|---|---|
| If your child has hearing problems, would you prefer to know this early?—Yes | 512 | 100.0 |
| Do you think it's important for all babies to be tested for hearing in the early days of life?—Yes | 512 | 100.0 |
| Would you be willing to pay for a hearing screening?—Yes | 512 | 100.0 |

example a study from Nigeria, where there is no national universal newborn hearing screening but neonatal hearing screening is offered in the frame of research projects, suggested that about 63% of mothers were not aware of neonatal hearing screening as well [15]. This finding suggests that greater work needs to be done so that prospective mothers and the general public are informed about the screening of newborn hearing and the places where this testing is offered. Research shows that better informed mothers have more positive attitudes toward universal newborn hearing screening programs [16].

A significantly higher percentage of babies born to mothers residing in Kukës and Pogradec, in rural areas, mothers with secondary or tertiary education, mothers with low or middle social status and mothers with very good/good health status were informed that newborn hearing screening was provided in the maternity hospital compared to their respective counterparts. These findings suggest that screening information efforts should target less informed groups.

On the other hand, about 80% of mothers in Tirana were not informed that hearing screening was provided in the two Tirana maternity hospitals. This result seems to be partly influenced by the fact that about 20% of births here occurred among mothers residing in other cities not being specifically covered by awareness campaigns for newborn hearing screening. Another explanation can be found in the much more dynamic, active and very heterogeneous population in Tirana compared to Kukës and Pogradec, where the latter are relatively small communities and the information is disseminated faster and more effectively than in Tirana. However, the finding regarding Tirana suggests that information campaigns in big cities should be different from those employed in small cities in terms of the aggressiveness of the coverage and the ways in which information is communicated.

About 67% of mothers informed about newborn hearing screening had received this information at the maternity ward after giving birth, by the relevant medical staff there, just as it occurs in most countries [16]. In Nigeria mothers learned about the availability of newborn hearing screening from antenatal clinics (28%), media (11%), family members or friends (17%), Internet (44%), etc. [15]. In Hong Kong about 69% of mothers learned about neonatal hearing screening at the postnatal ward premises and other information sources included previous screening of older children (16.9%), friends or relatives (5.9%) etc. [16]. It seems that in Albania, for the time being, the information sources other than the maternity hospitals where UNHS is taking place play a rather small role in informing the population about this screening program. On the other hand, international research suggests that most mothers would like to be informed about newborn hearing screening well in advance and probably not in maternity hospital premises [17]. The implication for the future is that there is need to improve and expand UNHS awareness campaigns in Albania to cover also mass media, social media and the Internet.

The overwhelming majority (94.1%) of mothers who gave birth to children in the maternities included in the EUSCREEN project in Albania were either very satisfied or satisfied with the hearing screening provided to their babies. This data suggests that the staff involved in hearing screening in all 4 maternities in Albania has performed the procedure properly, gaining the confidence of mothers. Parents' positive attitudes towards UNHS has also been reported in literature [16, 18]. Also, there was a positive relationship between satisfaction with newborn hearing screening with social class and economic status, in accordance with previous research [15].

On the other hand, a smaller percentage of mothers in Tirana are more satisfied with hearing screening compared to mothers who gave birth in Kukës or Pogradec. The implication is that measures should also be taken in Tirana to increase the percentage of mothers who are very satisfied/satisfied with the screening of their newborn baby. In this regard, efforts should be focused on mothers with low social status and disadvantaged economic status.

In our study about two-thirds of mothers (61.5%) were very stressful or stressful waiting for the hearing screening result. Being anxious about the screening results is also reported by previous research and it seems to be mainly associated with the inconclusive message (the result of the screening test is not definitive and there is need for further testing) at the end of each screening stage [19]. In addition, the role of screeners seems to be fundamental for reassuring parents about the need to be screened again [19]. The implications are that a better job should be done by the relevant hearing screening staff for supporting mothers and informing them about the screening procedure, stages and interpretation of results and intervention opportunities.

All mothers included in the current study (100%) stated that they would prefer to know early if their child has hearing problems. This finding best supports the value and importance of hearing screening in newborns as this procedure provides the fastest possible responses (meeting future mothers' expectations) and ensures early detection of the problem, greatly increasing the chances of early treatment, and potential full rehabilitation of infants affected by these conditions. The willingness to know early if own child has a hearing problem is supported by international research [16, 20]. Knowing early does not take away the grief that comes with the unfavorable diagnosis but it could give parents the needed time to somehow getting used to this situation and it can give them a sense of being able to take action quickly [20].

The importance of early detection and early treatment is also supported by the opinions of the mothers included in the study, where all mothers (100%) stated that it is very important for all babies to be tested for hearing in the early days of life, similar to international research results showing a very high level of parental agreement with regard to implementation and importance of newborn hearing screening [15, 18]. This is an additional argument in favor of universal hearing screening of all newborns, as this procedure is not only of medical and general population health benefit but it also enables the achievement of an aspiration, expectation or ideal of future mothers as an integral part of the universal right to good health.

As literature review shows, there are evidence-based arguments to support the implementation of UNHS [5]. However, early identification needs to be followed by timely fitting, amplification and family support for the affected babies; this requires effective follow-up of babies with hearing loss being detected by UNHS and often this represents the real challenge of the UNHS [7]. Research shows that on the one hand parents do support early identification of hearing loss but on the other hand they are rightly concerned about the effectiveness of follow-up and appropriate treatment for the affected babies [21]. Indeed, early identification [within the first 6 months of life] of hearing loss or impairment is accompanied by many different challenges for the parents, including timing, early amplification issues, information and support, daily management issues, etc. [21], issues that call for the development of services that appropriately address early identification of hearing impairment [21]. In addition, it should be kept in mind that most mothers have rather a limited understanding of their baby's hearing development milestones [16], which means that passing the newborn screening test is not a full assurance that future hearing problems will not occur and therefore mothers have to be able to continually monitor the baby's growth and detect any warning signs that something is not right with baby's hearing [16]. The relevant implication is that prospective mothers need to be better educated about child hearing development milestones as well.

All mothers included in the study (100%) stated that they would be willing to pay for their child's hearing medical control. This finding assures us once again about the public perception that universal newborn hearing screening is important, and that is why all possible efforts should be taken to achieve full coverage of all babies born in Albania. The finding specifically suggests that such activity is so positively perceived and highly appreciated by mothers as there

may be room for discussions about potential parental involvement in universal hearing screening funding schemes (in the event of an absolute inability to be 100% covered by public funds), economic contribution which undoubtedly needs to be tailored to the socio-economic level and other factors and based on the results of extensive consultations with all stakeholders.

### Study limitations

Our study has several limitations. The interviewing of consecutive mothers showing up to give birth in the selected maternity hospitals does not exclude the selection bias. However, we covered all the UNHS pilot sites in Albania and also selected the mothers proportional to size of respective maternities, in order to reduce the selection bias. On the other hand, the study sample is rather small and does not allow the unconditional generalization of the results. Finally, the cross-sectional nature of the study does not allow us to draw any causal conclusions.

The strong point of the present study is that it offers for the first time a glimpse on mothers' perception about newborn hearing screening, a largely under researched topic in Albania.

## Conclusions

In Albania mothers in general had positive attitudes toward universal newborn hearing screening program. The overwhelming majority of mothers giving birth to maternity hospitals where UNHS is being implemented were aware what their newborn is tested for, meaning the screening staff has done its job quite well. On the other hand, about two-thirds of mothers didn't know the hearing screening was offered in these maternity hospitals with considerable differences between the capital and two other regions, implying further targeted information efforts. About two-thirds of mothers were stressed or anxious waiting for the testing results. All mothers were in favor of early detection of hearing problems among newborn babies, they wished to know early if their baby has a hearing problem and they are willing to contribute financially for testing the hearing of their newborn babies. The various significant socio-demographic differences noticed regarding mother's awareness, information and satisfaction rates about various UNHS elements and perceived stress of screening results should guide information and education campaigns. The public willingness to financially contribute in order to screen the hearing of newborns might be an opportunity for scaling-up the UNHS in the country.

## Supporting information

**S1 Database.**
(SAV)

## Author Contributions

**Conceptualization:** Birkena Qirjazi, Ervin Toçi.

**Data curation:** Birkena Qirjazi, Ervin Toçi.

**Formal analysis:** Birkena Qirjazi, Ervin Toçi.

**Investigation:** Birkena Qirjazi, Ervin Toçi.

**Methodology:** Genc Burazeri, Enver Roshi.

**Writing – original draft:** Birkena Qirjazi, Ervin Toçi.

**Writing – review & editing:** Birkena Qirjazi, Ervin Toçi, Eduard Tushe, Genc Burazeri, Enver Roshi.

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
