## [Decision Letter · Decision Letter 0]

7 Jul 2020

PONE-D-20-07930

Mothers’ perceptions of universal newborn hearing screening in transitional Albania

PLOS ONE

Dear Dr. Toçi,

Thank you for submitting your manuscript to PLOS ONE. After careful consideration, we feel that it has merit but does not fully meet PLOS ONE’s publication criteria as it currently stands. Therefore, we invite you to submit a revised version of the manuscript that addresses the points raised during the review process.

We look forward to receiving your revised manuscript.

Kind regards,

Bolajoko O. Olusanya, MBBS, FMCPaed, FRCPCH, PhD

Academic Editor

PLOS ONE

Journal Requirements:

2.Thank you for including your ethics statement:

"All mothers agreed to participate after being thoroughly informed about the survey.

EUSCREEN project in Albania has been approved by the project counterpart in Albania, i.e. the University of Medicine, and the Ministry of Health and Social Protection of Albania."

a. Please amend your current ethics statement to confirm that your named institutional review board or ethics committee specifically approved this study.

5.Thank you for stating the following in the Acknowledgments Section of your manuscript:

'This project has received funding from the European Union’s Horizon 2020 research and innovation program under grant agreement No 733352.This study sponsor had no role in the study design, or the collection, analysis and interpretation of the data or in the writing of the report or in the decision to submit the article for publication.'

'The funders had no role in study design, data collection and analysis, decision to publish, or preparation of the manuscript'

6. Please include captions for your Supporting Information files at the end of your manuscript, and update any in-text citations to match accordingly. Please see our Supporting Information guidelines for more information: http://journals.plos.org/plosone/s/supporting-information

Reviewers' comments:

Reviewer's Responses to Questions

**Comments to the Author**

1. Is the manuscript technically sound, and do the data support the conclusions?

Reviewer #1: No

Reviewer #2: Yes

2. Has the statistical analysis been performed appropriately and rigorously? 

Reviewer #1: No

Reviewer #2: No

3. Have the authors made all data underlying the findings in their manuscript fully available?

Reviewer #1: Yes

Reviewer #2: Yes

4. Is the manuscript presented in an intelligible fashion and written in standard English?

Reviewer #1: No

Reviewer #2: Yes

5. Review Comments to the Author

Reviewer #1: A cross-sectional study was carried out to explore parents’ perceptions about various elements three sites of UNHS in Albania and included 512 mothers. While the study is a needed study as there is paucity of research from this region, there are several concerns.

1. The words 'mothers' and 'parents' are used interchangeably throughout the manuscript, making it difficult for a reader to derive conclusions.

2. The questionnaire development does not follow standard procedures for questionnaire development and validation. There is no mention who were these 'local experts'?Also no mention of the what was this 'appropriate literature' challenges the overall validity of the questionnaire and thereby the methodology. The usage of terms like 'thought to have' should be avoided in scientific writing.

3. The study was approved by Ministry of Health and Social Protection and University of Medicine, however there is no mention of Institutional committee approval. Also why was oral consent considered valid? Shouldnt a written informed consent be taken from all the participants.

4. The analysis was restricted to Chi-square test, and no detailed statistical exploration was carried out.

5. Were any standard checklists used to determine the general health status as well as economic status?

6. Clinical implications, future directions and limitations could have been added to enrich the discussion further.

7 Overall, the manuscript would benefit if read and corrected by a native English language expert in scientific writing.

Reviewer #2: 1.The manuscript is technically sound, and the data support the conclusions . However the conclusion should be precise and reflect the study objective.

2.Statistical analysis were performed but the author should use logistic regression to aid determination of factors that were most likely responsible for perception of mothers.

3. The authors made all data underlying the findings in their manuscript fully available.

4.The manuscript was presented in an intelligible fashion and written in standard English however the word stress/stressed used need to be clarified to avoid ambiguity.

6. PLOS authors have the option to publish the peer review history of their article (what does this mean?). If published, this will include your full peer review and any attached files.

Reviewer #1: No

Reviewer #2: **Yes: **Abayomi Oladapo Somefun

---

## [Author Response · Author response to Decision Letter 0]

8 Jul 2020

Response to Reviewers

Manuscript PONE-D-20-07930: “Mothers’ perceptions of universal newborn hearing screening in transitional Albania” 

We thank the reviewers for their constructive comments which have helped us to improve the content and format of the manuscript, which we hope is now acceptable for publication in the PLOS ONE Journal. 

Below, we address point-by-point all the reviewers’ comments:

Reviewer #1: 

1. “The words 'mothers' and 'parents' are used interchangeably throughout the manuscript, making it difficult for a reader to derive conclusions.”

Response: We thank the reviewer for this valuable comment and suggestion. In line with reviewer’s suggestion, we decided to keep only the word “mother’s” and therefore in the revised manuscript we have now only used the terms “mother’s”, where appropriate. 

2. “The questionnaire development does not follow standard procedures for questionnaire development and validation. There is no mention who were these 'local experts'? Also no mention of the what was this 'appropriate literature' challenges the overall validity of the questionnaire and thereby the methodology. The usage of terms like 'thought to have' should be avoided in scientific writing”

Response: Point well-taken. In the revised manuscript, we have now reformulated the relevant section in order to reflect the concerns and suggestions of the reviewer, so that now this section reads like this: “The questionnaire used for this survey has been built from the local experts (the authors of this paper), based on browsing of the appropriate literature listed in the references and picking up those items considered as relevant for the study objectives. Therefore, the questionnaire has face validity, construct validity and content validity.”

3a. “The study was approved by Ministry of Health and Social Protection and University of Medicine, however there is no mention of Institutional committee approval. 

Response: We thank the reviewer for bringing this point into our attention. We are trying to explain again the situation regarding this point: The EUSCREEN in Albania has been approved by the Ministry of Health and Social Protection (MoHSP) and the implementing partner, the University of Medicine. We (the project coordinator in Albania) have the document stating such an agreement and approval and, if the reviewer thinks it necessary, we can make this document available to him/her or to any other interested party. This approval is signed by the Minister of Health and Social Protection and the Rector of the University of Medicine. The Bio-ethical Committee that is functioning in Albania is dependent on the MoHSP and therefore, the agreement made by the Minister of Health has covered the issue of the Bio-ethical Committee as well. This approval includes all data collection process in the framework of EUSCREEN study in Albania, including hearing screening tests and other questionnaires for the mothers and or staff. For example, in the framework of EUSCREEN data collection in Albania, we ask the mothers also about various risk factors for hearing impairment (such as family history, genetic disorders, use of antibiotics during pregnancy etc.), and many other socio-demographic factors (age, education, residence, financial situation, etc.). Other investigators from the EUSCREEN consortium have also come to Albania and interviewed the staff for their purposes, and they have even published papers based on that information and have stated exactly the approval given by the MoHSP and the agreement between MoHSP and the University of Medicine as a proof of ethical approval of the study (reference: Bussé AM, Qirjazi B, Goedegebure A, et al. Implementation of a neonatal hearing screening programme in three provinces in Albania. Int J Pediatr Otorhinolaryngol. 2020;134:110039. doi:10.1016/j.ijporl.2020.110039). In this context, the data collection made for the purposes of this study is in total agreement with the approval given by the MoHSP and following the data collection in the framework of EUSCREEN in Albania.

In the manuscript we have explained this situation with the following paragraph: “The UNHS project in Albania was approved by Ministry of Health and Social Protection, in collaboration with the implementing partner, the University of Medicine. This approval comprised all elements of screening procedure, including various data collection activities from the mothers (i.e. demographic and socioeconomic questionnaire, parents’ opinion on newborn hearing screening, reasons for refusing screening, etc.).” 

3b. “Also why was oral consent considered valid? Shouldnt a written informed consent be taken from all the participants.”

Response: Point well-taken. In order to be part of the newborn screening, each mother has to give a written informed consent, which is administered by the hospital staff. If such written consent was not given, the mother and the baby could not be part of the UNHS. Among mothes giving their written consent to be part of the screening process, for every other survey we formally asked also for the oral consent of the mothers.

In order for this to be clear, we have now reformulated the respective section in the manuscript, so as now it reads like this: “In order to be part of hearing screening, each mother was required to give a written informed consent. If such consent was not retrieved, then the newborn was not screened, the mother’s interviewing was not done and this particular baby was not part of UNHS anymore. All mothers accepting to screen their baby were required to give a formal oral consent for participating in the actual study.” 

4. “The analysis was restricted to Chi-square test, and no detailed statistical exploration was carried out”

Response: We thank the reviewer for this comment. Actually, the data analysis was guided by the nature of the variables in our disposal. We kindly remind the reviewer that this is a small mainly descriptive study that does not aim to engage into complex statistical models or analysis but it rather tries to give an overall picture of mother’s perceptions about UNHS in Albania. On the other hand, the other papers from international area also have reported simple analytical tools in order to give such a picture. At the end, we did not aim to detect the predictors or factors related to various mothers’ perceptions and therefore the chi square test was regarded as an optimal test to check for the present differences in our case.

5. “Were any standard checklists used to determine the general health status as well as economic status?”

Response: We thank the reviewer for this comment. Actually, the general health status and the economic status of the mothers was self-reported, based on the well-recognized questions: “How would you rate your general health status? (ranging from 1 (Very good) to 5 (very poor)” and “On a scale from1 (very poor) to 5 (not poor), how do you consider yourself?” 

6. “Clinical implications, future directions and limitations could have been added to enrich the discussion further”

Response: We thank the reviewer for this suggestion. In the revised manuscript we have now made some additional efforts to address the points raised by the reviewer. The added paragraphs are listed below:

“On the other hand, international research suggests that most mothers would like to be informed about newborn hearing screening well in advance and probably not in maternity hospital premises [17]. The implication is that there is need to improve and expand UNHS awareness campaigns to cover also mass media, social media and the Internet”

“In addition, it should be kept in mind that most mothers have rather a limited understanding of their baby’s hearing development milestones [16], which means that passing the newborn screening test is not a full assurance that future hearing problems will not occur and therefore mothers have to be able to continually monitor the baby’s growth and detect any warning signs that something is not right with baby’s hearing [16]. The relevant implication is that prospective mothers need to be better educated about child hearing development milestones as well.”

In line with reviewer’s suggestion, we have now added a “Study limitations” section, which in the revised manuscript reads like this: 

“Our study has several limitations. The interviewing of consecutive mothers showing up to give birth in the selected maternity hospitals does not exclude the selection bias. However, we covered all the UNHS pilot sites in Albania and also selected the mother’s proportional to size, in order to reduce the selection bias. On the other hand, the study sample is rather small and does not allow the unconditional generalization of the results. Finally, the cross-sectional nature of the study does not allow us to draw any causal conclusions.

The strong point of the present study is that it offers for the first time a glimpse on mothers’ perception about newborn hearing screening, an under researched topic in Albania.”

7. “Overall, the manuscript would benefit if read and corrected by a native English language expert in scientific writing”

Response: We thank the reviewer for this suggestion. We have now revised the English language of the manuscript and have done all the necessary changes in order to comply with this suggestion. The changes are marked throughout the manuscript, where necessary. 

Reviewer #2: 

1. “The manuscript is technically sound, and the data support the conclusions . However the conclusion should be precise and reflect the study objective.”

Response: We thank the reviewer for this valuable comment and suggestion. We have gone through our results and discussion and feel that in general they are in concordance with each-other; we have tried to structure the discussion following our results. In addition, we have now expanded the discussion as to include some other relevant topics and have also added the “Study limitation” section. We do hope that these changes will satisfy the reviewer’s concern about this point.

2. “Statistical analysis were performed but the author should use logistic regression to aid determination of factors that were most likely responsible for perception of mothers.”

Response: We thank the reviewer for this comment. Actually, the data analysis was guided by the nature of the variables in our disposal. We kindly remind the reviewer that this is a small mainly descriptive study that does not aim to engage into complex statistical models or analysis but it rather tries to give an overall picture of mother’s perceptions about UNHS in Albania. On the other hand, the other papers from international area also have reported simple analytical tools in order to give such a picture. At the end, we did not aim to detect the predictors or factors related to various mothers’ perceptions and therefore the chi square test was regarded as an optimal test to check for the present differences in our case.

3. “The authors made all data underlying the findings in their manuscript fully available.”

Response: We agree.

3. “The manuscript was presented in an intelligible fashion and written in standard English however the word stress/stressed used need to be clarified to avoid ambiguity.”

Response: We thank the reviewer for this suggestion. We have now added the word “anxious” next to the word “stressed” in order to not confuse the reader. In the revised manuscript, the only sentence where the word “stressed” is used, now reads like this “About two-thirds of mothers were stressed or anxious waiting for the testing results.”

In the other circumstances where the word “stress” is used it is followed by an explanation that this refers to the anxiety waiting for the hearing screening results. For example, in the manuscript, lines 294-298 state that “In our study about two-thirds of mothers (61.5%) were very stressful or stressful waiting for the hearing screening result. Being anxious about the screening results is also reported by previous research and it seems to be mainly associated with the inconclusive message (the result of the screening test is not definitive and there is need for further testing) at the end of each screening stage [18]”. To our opinion, the reader is clear enough that the stress here refers to the anxiety related to waiting for the hearing screening results.

---

## [Decision Letter · Decision Letter 1]

4 Aug 2020

Mothers’ perceptions of universal newborn hearing screening in transitional Albania

PONE-D-20-07930R1

Dear Dr. Toçi,

We’re pleased to inform you that your manuscript has been judged scientifically suitable for publication and will be formally accepted for publication once it meets all outstanding technical requirements.

Kind regards,

Bolajoko O. Olusanya, MBBS, FMCPaed, FRCPCH, PhD

Academic Editor

PLOS ONE

Additional Editor Comments (optional):

Reviewers' comments:

Reviewer's Responses to Questions

**Comments to the Author**

1. If the authors have adequately addressed your comments raised in a previous round of review and you feel that this manuscript is now acceptable for publication, you may indicate that here to bypass the “Comments to the Author” section, enter your conflict of interest statement in the “Confidential to Editor” section, and submit your "Accept" recommendation.

Reviewer #1: All comments have been addressed

2. Is the manuscript technically sound, and do the data support the conclusions?

Reviewer #1: Partly

3. Has the statistical analysis been performed appropriately and rigorously? 

Reviewer #1: Yes

4. Have the authors made all data underlying the findings in their manuscript fully available?

Reviewer #1: Yes

5. Is the manuscript presented in an intelligible fashion and written in standard English?

Reviewer #1: Yes

6. Review Comments to the Author

Reviewer #1: Satisfactory changes and justification done. However, i suggestion to separate the questionnaire into the demographic details part and the questionnaire aspect

7. PLOS authors have the option to publish the peer review history of their article (what does this mean?). If published, this will include your full peer review and any attached files.

Reviewer #1: No

---

## [Editor Report · Acceptance letter]

11 Aug 2020

PONE-D-20-07930R1 

Mothers’ perceptions of universal newborn hearing screening in transitional Albania 

Dear Dr. Toçi:

I'm pleased to inform you that your manuscript has been deemed suitable for publication in PLOS ONE. Congratulations! Your manuscript is now with our production department. 

Kind regards, 

on behalf of

Dr. Bolajoko O. Olusanya 

Academic Editor

PLOS ONE